# Hydrostatic Filtration Enables Large-Scale Production of Outer Membrane Vesicles That Effectively Protect Chickens against *Gallibacterium anatis*

**DOI:** 10.3390/vaccines8010040

**Published:** 2020-01-23

**Authors:** Fabio Antenucci, Homa Arak, Jianyang Gao, Toloe Allahgadry, Ida Thøfner, Anders Miki Bojesen

**Affiliations:** Department of Veterinary and Animal Sciences, University of Copenhagen, DK-1870 Frederiksberg C, Denmark; fante@sund.ku.dk (F.A.); homaarak@gmail.com (H.A.); 648291851a@gmail.com (J.G.); toloe@sund.ku.dk (T.A.); icnt@sund.ku.dk (I.T.)

**Keywords:** *Gallibacterium anatis*, outer membrane vesicles, vaccination, hydrostatic filtration

## Abstract

*Gallibacterium anatis* is a Gram-negative opportunistic avian pathogen representing an emerging threat to poultry meat and egg production worldwide. To date, no vaccine able to effectively prevent the morbidity associated with *G. anatis* infections has been developed yet. Our group previously reported that inoculation of different combinations of *G. anatis* outer membrane vesicles (OMVs), FlfA and GtxA-N proteins is effective in preventing lesions caused by *G. anatis* infections in layer chickens. Here we report the testing of the efficacy as vaccine prototypes of *G. anatis* OMVs isolated by hydrostatic filtration, a simple technique that allows the cost-effective isolation of high yields of OMVs. Layer chickens were immunized with OMVs alone or in combination with FlfA and/or GtxA-N proteins. Subsequent challenge with a heterologous *G. anatis* strain showed that immunization with OMVs alone could significantly reduce the lesions following a *G. anatis* infection. A second study was carried out to characterize the dose-response (0.25, 2.5 and 25 µg) relationship of *G. anatis* OMVs as immunogens, showing that 2.5 μg of OMVs represent the optimal dose to elicit protection in the immunized animals after a similar challenge. Additionally, administration of ≥2.5 μg of *G. anatis* OMVs induced specific IgY titers and possibly vertical transfer of immunity.

## 1. Introduction

*Gallibacterium anatis* is a Gram-negative bacterium belonging to the Pasteurellaceae family. Although commonly present in healthy birds, most *G. anatis* isolates can cause opportunistic infections in both domesticated and non-domesticated birds [1,2]. *G. anatis* infections can result in a wide variety of pathological manifestations, including septicemia, pericarditis, hepatitis, respiratory tract lesions and enteritis, but are most commonly associated with the development of salpingitis, acute peritonitis and hemorrhagic/ruptured follicles [2,3,4,5].

In affected animals, *G. anatis* is often co-isolated together with other poultry pathogens, which previously left its clinical relevance as a matter of controversy [4,6]. Several studies have however shown that *G. anatis* infections correlate with decreased egg production and increased mortality in laying hens [7,8,9], leading to the widespread recognition of *G. anatis* as an emerging threat to chicken egg and meat production.

Although some inactivated vaccines against *G. anatis* have been developed with a certain degree of success [10], there is still no available vaccine on the market able to confer protection against the most prevalent *G. anatis* biovars and strains. This has been due in part to the high genetic variation found among *G. anatis* isolates [11,12]; a factor that significantly complicated the identification of the conserved immunogens required for the development of a vaccine offering broad coverage. In this respect, outer membrane vesicles (OMVs), a type of vesicles secreted from the outer membrane (OM) of the majority of Gram-negative bacteria have shown promising perspectives [13]. Due to the nature of OMVs biogenesis, the vesicles usually possess an antigenic pattern similar to the one found on the parental OM [13,14]. This feature leaves OMVs as ideal candidates for the development of novel vaccines against Gram-negative pathogens [15], a fact that has been proven by the development and commercialization of an effective OMV-based vaccine against *Neisseria meningitidis* for human use [16]. Additionally, the OMVs have been shown to act as a potent adjuvant when co-administered with other immunogens [17,18,19], further increasing the possible applications of vesicles in vaccine development [20]. Our group previously reported that *G. anatis* secretes OMVs [21], and that these vesicles, in combination with the conserved proteins FlfA or the N-terminal of the GtxA toxin [22,23,24], are effective in preventing the development of lesions associated with *G. anatis* infections when administered as immunogens [18,25]. FlfA is a conserved fimbrial protein involved in *G. anatis* adhesion and pathogenesis. FlfA homologs are expressed by a wide range of *G. anatis* strains, underlining the potential of this protein as a cross-serovar immunogen [22]. GtxA is a large secreted toxin belonging to the repeat-in-toxin (RTX) family of toxins [26]. The leukotoxic and hemolytic activity exhibited by GtxA in vitro have been hypothesized to represent a central determinant of pathogenesis of hemolytic *G. anatis* strains [27,28].

One of the factors that can determine the feasibility of translating an experimental vaccine to a commercial product is the ability to upscale production in a standardized and cost-effective manner. While OMVs have traditionally been isolated by ultracentrifugation, gradient centrifugation, ultrafiltration or size exclusion chromatography [29], these techniques present either relative high operative costs or lack the reliability, yields and reproducibility needed for large-scale production. Instead, our focus has been to isolate *G. anatis* OMVs by hydrostatic filtration (HF), an effective and simple technique originally reported by Musante et al. [30], which we successfully adapted and employed for the large-scale isolation of OMVs from the Gram-negative pathogen *Actinobacillus pleuropneumoniae* [17,31].

In the current investigation we aimed at providing a thorough characterization of *G. anatis* OMVs produced by HF in the context of vaccine development. Accordingly, the objectives were the following: (i) cost-effective large-scale production and characterization of *G. anatis* OMVs isolated by HF; (ii) determination of the most effective combination of OMVs, FlfA and GtxA-N proteins as vaccine formulation against an experimental challenge with live *G. anatis* cells; (iii) determination of the dose/response relationship of the selected vaccine formulation and (iv) characterization of the serological response elicited against *G. anatis* OMVs following immunization of the chicken layers.

## 2. Materials and Methods

### 2.1. Immunogens Production

The recombinant proteins FlfA and GtxA-N employed in this study were previously produced as described in [32]. GtxA-N represents the first 949 amino acids on the N-terminal of the GtxA toxin expressed by *G. anatis* 12656-12. OMVs were produced from the hyper-vesiculating *G. anatis* 12656-12 Δt*olR* mutant [21] and subsequently isolated using a modified version of the HF procedure described in [31]. Briefly, 300 mL of pre-heated (37 °C) brain–heart infusion (BHI; Gibco) broth was inoculated with 3 mL of an overnight (ON) culture (1% *v*/*v*) and incubated until late exponential phase (37 °C/200 rpm). Culture supernatants were obtained by centrifugation (5000*g*, 15 min, 4 °C) and syringe-filtered through 0.45 μM filters (Sartorius, Göttingen, Germany) to produce cell-free crude supernatant isolates. The crude isolates were loaded into a cellulose ester dialysis membrane with pore-size range of 1000 kDa (Repligen, Waltham, MA, USA), concentrated by HF [31] and dialyzed twice 1:100 (4 h, 4 °C) in sterile phosphate buffered saline (PBS; Sigma-Aldrich, St. Louis, MO, USA) to eliminate supernatant contaminants. An additional concentration to a final volume of 3 mL OMV containing filtrate was performed using Amicon Ultra-15 Centrifugal Filter Units with Ultracel-10 membrane (Merck Millipore, Burlington, MA, USA). The concentrated OMVs batches produced were then aliquoted and stored at −20 °C. A total of two batches of *G. anatis* OMVs were produced for this study.

### 2.2. Characterization of OMV Batches

The OMV batches produced were characterized by: (i) SDS-PAGE (sodium dodecyl sulphate—polyacrylamide gel electrophoresis) analysis; (ii) cryo transmission electron microscopy (Cryo-TEM) imaging and (iii) tunable resistive pulse sensing (TRPS) quantification.

(i)Different OMV concentrations were loaded on 10% NuPAGE^®^ Novex^®^ Bis–Tris gels (Invitrogen, Carlsbad, CA, USA) and run under reducing conditions. PageRulerTM Plus Prestained Protein Ladder (Thermofisher Scientific, Waltham, MA, USA) was used for size determination. Protein gels were then stained with SimplyBlueTM SafeStain (Invitrogen). Protein concentration was qualitatively estimated by comparison with a protein standard of known concentration (not shown).(ii)Of the nanodispersed OMVs 3 µL was applied on a hydrophilized lacey carbon 300 mesh copper grid (Ted Pella Inc., Redding, CA, USA). The excess sample on the grid was then blotted with filter paper at a blotting time of 3 s, blotting force 0, temperature 4 °C and 100% humidity (FEI Vitrobot IV), and was rapidly plunged into liquid nitrogen cooled ethane (−180 °C). Sample observations were performed using a Tecnai G2 20 transmission electron microscope (FEI) at a voltage of 200 kV under a low-dose rate. Images were recorded with an FEI Eagle camera 4 k × 4 k at variable nominal magnifications.(iii)OMVs batches were quantified by TRPS using a qNano device (Izon Sciences Ltd., Christchurch, New Zealand) following a standard protocol [33]. Data were analyzed using the data capture and analysis software, Izon Control Suite V.3.3.2.2001.

### 2.3. In Vivo Studies

Two separate in vivo studies were carried out in order to determine: (i) which vaccine formulations could provide the best protection against morbidity caused by *G. anatis* infections (study 1) and (ii) the dose-response effect of the concentration of OMVs administered on the clinical outcome of infection and the development of specific antibody titers (study 2). The design of studies 1 and 2 is summarized in Figure 1. Selected time points in the studies were: T0 (day 0, first immunization); T1 (day 28, second immunization); T2 (day 42, challenge) and T3 (day 49, necropsy). 

The immunization regimes presented in study 1 include all possible combinations of *G. anatis* OMVs and the recombinant proteins FlfA and GtxA-N (Figure 1). Although several of these immunogen combinations have been previously tested as vaccine candidates [18,22,23,24,25], a systematic study including all possible combinations of these immunogens and HF-isolated *G. anatis* OMVs was not reported yet. 

### 2.4. Animal Model

Two batches of 120 and 80 16-weeks old Lohmann-Brown layer chickens were from a Danish commercial rearing farm with high biosecurity level as defined by Bojesen and coworkers [34]. Upon arrival, the animals were housed at the Department of Veterinary and Animal Sciences, (Frederiksberg, Copenhagen, Denmark). The chickens were leg-tagged with a unique identification number and randomly distributed in six separate pens (20 animals/pen). Experiments were started after a one-week acclimatization period. Daily care was provided by animal caretakers blinded to treatment groups. Experiments were approved by the Danish national animal experiments inspectorate (Dyreforsøgstilsynet), as stipulated in license numbers (2013-15-2934-00923 and 2019-15-0201-01611). 

All animal experiments detailed in this study comply with the Animal Research: Reporting of In Vivo Experiments (ARRIVE) guidelines and EU Directive 2010/63/EU for animal experiments.

### 2.5. Immunization

Immunization regimes adopted in study 1 and 2 are summarized in Figure 1. After one-week acclimatization the layer chickens were immunized intra-muscularly (IM) with either of the recombinant proteins FlfA and GtxA-N, OMVs or a combination of OMVs and recombinant proteins (Figure 1). Animals from group 1 (controls) were immunized IM with a mock inoculum of sterile soluble protein buffer (50 NaP, 150 NaCl, 0.5 mM TCEP and 10% glycerol; pH 7.5). All the immunogens employed in the immunization procedures were drawn from the same batches of OMVs (batch 1), FlfA and GtxA-N proteins. The inocula containing immunogens were prepared by resuspending the immunogens in sterile soluble protein buffer to the desired concentration. Immunization doses were prepared by loading 0.5 mL of the inocula in 1 mL disposable syringes (Kruuse), while immunizations were performed by injection in the breast muscle. Four weeks after first immunization the animals received a booster immunization of identical dose and formulation. Blood for serum separation was drawn from the brachial vein on days 0, 14, 28, 35, 42 and 49.

### 2.6. Challenge

Challenge regimes adopted in study 1 and 2 are summarized in Figure 1. Two weeks after second immunization the chickens were challenged with an intraperitoneal (IP) injection of 1 × 10^7^ (study 1) or 5 × 10^7^ (study 2) live colony-forming units (CFU) of *G. anatis* 10672/9. Strain 10672/9 belongs to a different serotype than strain 12656-12 and was used to test the breath of protection induced by the vaccine formulation (heterologous challenge; Figure 1). Challenge inocula were prepared as follows. 200 mL of pre-heated (37 °C) BHI broth was inoculated with 2 mL of an ON culture of *G. anatis* 10672/9 (1% *v*/*v*) and incubated until reaching optical density_600_ (OD_600_) 1.5 (37 °C/200 rpm), corresponding to 2 × 10^9^ CFU/mL. Cultures were then serially diluted (1:10) in sterile PBS to obtain the desired bacterial concentrations in a final inoculation volume of 0.5 mL. Challenge doses were prepared by loading 0.5 mL of the diluted cultures in 1mL disposable syringes. Loaded syringes were stored at 4°C during the inoculation procedure. Serial dilutions of the inocula before (pre) and after (post) challenge were spotted and incubated (37 °C/ON) in sextuplicates on BHI-agar plates supplemented with 5% *v*/*v* calf blood for CFU counting the following day. CFU counts pre and post-challenge were statistically compared using the Wilcoxon matched-pairs signed rank test.

### 2.7. Post-Mortem Examination

One week after the challenge the chickens were sacrificed and subjected to necropsy. Peritoneum, ovary, salpinx and spleen were examined to determine the clinical outcome of infection. Parameters such as inflammatory reaction, presence of exudate, vascularization, transparency of peritoneum and follicle rupture/regression were used to assign a macroscopic lesion score to each organ according to a pre-developed scoring system (Appendix A) [24], with a score of 0 representing the absence of detectable lesions. Lesion score ranges were the following: peritoneum (0–15); ovary (0–9); salpinx (0–9); spleen (0–5) and overall (0–38). Lesion scores of each group plotted by organ and as overall scores were statistically analyzed by Kruskal–Wallis (*p* < 0.05) and Mann–Whitney (*p* < 0.05) tests to assess significance.

### 2.8. Serum Preparation

Blood samples for serum extraction were left at room temperature for one day and at 4 °C for an additional day where the serum was extracted, aliquoted and stored at −20 °C.

### 2.9. IgY Extraction from Egg Yolk

All eggs laid by the layer chickens part of study 2 were collected and stored (4 °C). Eggs from day 28 (2nd immunization), 42 (challenge) and 49 (necropsy) were selected as representative time points. IgY were extracted from egg yolks by polyethylene glycol (PEG) precipitation according to the protocol described in [35]. The concentration of the extracted yolk IgY was tested by SDS-PAGE (data not shown). Yolk IgY samples were aliquoted and stored at −20 °C.

### 2.10. Antibody Titration

Specific IgY titers in sera and egg yolks were determined by enzyme-linked immunosorbent assay (ELISA). Unless otherwise specified, buffers, reagents and concentrations are similar to what is described in [17]. All tests were carried out in duplicates.

Serum: Nunc Maxisorp flat-bottom 96-wells plates (Thermo Fisher Scientific) were coated with 100 ng of FlfA and GtxA-N proteins, or 1:500 dilutions of OMVs in PBS, respectively. Pooled sera from animals in the same group were added in three-fold dilutions between 1:500 and 1:1,093,500 (FlfA, GtxA-N) or 1:1000 and 1:2,187,000 (OMVs).

Yolk IgY: Nunc Maxisorp flat-bottom 96-wells plates were coated with 1:500 dilutions of OMV isolates in PBS. Pooled yolk IgY samples from animals in the same group were added in three-fold dilutions between 1:4000 and 1:8,748,000.

All samples: IgY specifically recognizing FlfA, GtxA-N and OMVs were detected with horseradish peroxidase (HRP)-conjugated goat anti-Chicken IgY (H + L; Thermo Fisher Scientific). Wells were revealed using TMB PLUS2 (Kem-En-Tec Diagnostics). Optical density was measured at 450 nm using an ELISA plate reader (VersaMax Molecular Devices). IgY titers are expressed as the area under the curve (AUC) of the titration curves. AUC values on selected time points were compared and statistically analyzed by ordinary one-way ANOVA (*p* < 0.05) and Dunnett’s multiple comparison tests (alpha = 0.05) to assess significance. The development of IgY titers over time was tested by analyzing AUC values within each group (all time points) by ordinary one-way ANOVA (*p* < 0.05) and Dunnett’s multiple comparison tests (alpha = 0.05). Statistical significance of differences in AUC values between groups on selected time points was assumed only for groups (other than controls) that also showed significant IgY titers development over time. 

## 3. Results

### 3.1. OMV Batches Analysis

The results of the characterization of the OMV batches produced are presented in Figure 2. OMV batches 1 and 2 presented comparable protein concentration (Figure 2A), TEM profile (Figure 2B,C) and particle size and number (Figure 2D). Protein concentration in OMV batches 1 and 2 was estimated to be of 7–7.5 mg/batch.

### 3.2. Concentration of Challenge Inocula

CFU counts from plated aliquots of challenge inocula are summarized in Table 1. All batches of inocula showed concentrations of *G. anatis* cells compatible with the expected challenge dose. No significant mortality of *G. anatis* cells was observed during storage of the inocula between pre-challenge and post-challenge time points (2 h, 4 °C). 

### 3.3. Pathological Analysis

Total and ovary lesion scores from the animals involved in study 1 and 2 are presented in Figure 3. Individual lesion scores of salpinx and spleen are omitted in Figure 3 as no significant differences between the scores of different groups were observed.

Affected organs showed lesions coherent with *G. anatis* infections, such as peritonitis, purulent inflammation and follicle rupture/regression (Figure 4).

Study 1: Animal groups that received OMV-based immunizations (groups 3–6) presented significantly different total and ovary lesion scores, respectively, as compared to the control group (group 1). On the contrary, immunization with FlfA and GtxA-N alone (group 2) failed instead to produce any observable effect on lesion scores when compared to the control group.Study 2: Animals that were immunized with 2.5 μg OMVs (group 3) presented significantly different peritoneum, ovary and total lesion scores as compared to the unprotected control group (group 1). Animals immunized with 25 μg OMVs (group 4) only differed significantly from the control group in the peritoneum and total lesion scores. Immunization with 0.25 μg OMVs (group 2) failed to produce any observable reduction in lesion scores as compared to the control group.

### 3.4. Antibody Response to Immunization

IgY titers from sera and egg yolks are shown in Figure 5. Unless otherwise specified, the results presented in this section refer to the time of challenge. Sera from animals belonging to the same group were pooled together for immunological analysis. Accordingly, results presented in this section must be considered as group trends and not representative of individual animal responses.

Study 1: Immunization with *G. anatis* OMVs (groups 3–6) induced significantly higher OMV-specific IgY titers, even after first immunization (T1), than the controls. None of the animals immunized with FlfA however developed any detectable FlfA-specific IgY titers. Immunization with GtxA-N (groups 2, 5 and 6) induced significantly increased GtxA-N-specific IgY titers, although not until the second immunization was completed (T2).Study 2: Immunization with 2.5 μg and 25 μg OMVs (group 3 and 4, respectively) induced significantly increased OMV-specific serum IgY titers of immunized animals. Similar titers were also detected in pooled yolks from the eggs laid by animals immunized with 2.5 μg and 25 μg OMVs. The lower immunization dose of 0.25 μg (group 3) did not appear to induce detectable serum or yolk specific IgY titers.

## 4. Discussion

The last 15 years have seen an increasing interest in the diagnostic and therapeutic potential offered by the OMVs in human and veterinary medicine alike [15,16,18,36,37]. Despite the relative ease with which OMV can be purified though, one of the main challenges for the utilization of OMVs as therapeutic tools has been the ability of isolating large batches of high-purity OMVs in a cost-effective and standardizable manner. This challenge is in fact two-fold: (i) production of large amounts of OMVs in one batch; (ii) production, isolation and characterization of OMV batches in a cost-effective and standardized manner.

For *G. anatis*, our group has previously reported high OMV yields by the production of the hyper-vesiculating *G. anatis* 12656-12 Δt*olR* mutant as compared to the wild type strain [21]. The composition of OMVs originating from mutant and the wild type appear highly similar although some quantitative differences among the individual proteins may occur [21]. Concerning standardization, the data presented in the present study showed that the application of the HF protocol to the purification of *G. anatis* OMVs yielded two OMV batches significantly similar in protein concentration (Figure 2) and particle concentration, size and morphology (Figure 2B–D). Interestingly, no large OMVs (diameter > 150 nm) or OMV aggregates could be observed in the OMV batches produced in this study (Figure 2B,C), indicating that what previously described for *G. anatis* OMVs [21] could represent artifacts produced by the high centrifugal forces employed during OMV purification by ultracentrifugation.

Determination of the cost-effectiveness of OMV isolation by HF can be made by combining the data obtained from the OMV batch characterization with the experimental design adopted in the current study. Both OMV batches produced contained an estimated 7–7.5 mg of OMV proteins per batch (Figure 2A). Assuming a therapeutic dose of 5 μg/animal, the OMV yields obtained in batches 1 and 2 correspond to 6800–7000 doses/batch, with an operative cost of the HF device estimated at $150–200 (USD)/batch. Accordingly, we argued that production costs, yields, complexity and reproducibility associated with the HF protocol presented in this study are considerably more feasible than those previously reported in scientific literature in relation to isolation and purification of similar OMV yields [29].

The results from study 1 showed that immunization of the chicken layers with OMVs alone and OMVs + FlfA + GtxA-N provide the best overall protection against heterologous *G. anatis* challenge (Figure 3A). Interestingly, animals immunized with OMVs alone and OMVs + FlfA showed significantly more severe peritoneum lesions than animals immunized with OMVs + GtxA-N and OMVs + FlfA + GtxA-N, suggesting the effectiveness of GtxA-N as an immunogen in reducing lesions at the site of challenge. Immunization with unadjuvanted FlfA and GtxA-N failed to confer the animals any detectable protection (Figure 3A). This vaccine formulation was adopted in study 1 in order to compare the response elicited by immunizing solely with FlfA + GtxA-N against the response elicited when *G. anatis* OMVs were also added to the vaccine formulation. Given that immunization with both FlfA [18,22] and GtxA-N [23] was previously reported to be effective in providing protection when co-administered with *G. anatis* OMVs or other adjuvants, and considering the well known adjuvant role exerted by the OMVs [17,20], the lack of protection observed in group 2 can most likely be attributed to the absence of an adjuvant in the vaccine formulation FlfA + GtxA-N.

Animals immunized with any of the vaccine formulations containing OMVs developed significant specific IgY titers against the OMVs, even after first immunization (Figure 5A). This was in accordance to what was previously reported [38,39] and by our group [17,18,25], and was not surprising when considering the multi-antigenic nature and lipopolysaccharide (LPS) content of the OMVs. The animals that received any vaccine formulation containing GtxA-N presented specific IgY titers against this protein, although only after the second immunization (Figure 5A). Co-administration of GtxA-N with OMVs increased specific IgY titers against GtxA-N, underlining that *G. anatis* OMVs exert a similar adjuvant effect to what previously described for OMVs of other bacteria [17,18,19]. Notably, administration of FlfA + GtxA-N induced significant specific IgY titers against GtxA-N in the animals. This was in direct contrast with previously published results where co-administration of an adjuvant was found critical for eliciting a GtxA-N-specific IgY titer [18]. The fact that none of the groups immunized with vaccine formulations containing FlfA developed any significant FlfA-specific IgY titer confirmed that immunization with recombinant FlfA in the absence of an appropriate adjuvant is unable to elicit IgY titers against FlfA, as previously reported by our group [18].

Due to the similarity in the overall protection conferred by immunization with OMVs alone vs. OMVs + FlfA + GtxA-N, the vaccine formulation containing OMVs alone was selected for a dose/response study (study 2). This decision was partly driven by the need of reducing production costs in order to deliver not only an effective, but also an economically viable vaccine. The production of recombinant proteins is considerably more expensive than production of *G. anatis* OMVs by the HF protocol adopted in this study.

The results from study 2 showed that the immunization dose of 2.5 μg OMVs was the most effective in providing overall protection against heterologous *G. anatis* challenge (Figure 3B). Albeit a similar degree of overall protection was observed in animals immunized with 25 μg OMVs, this dosage notably failed to provide significant protection where most relevant, namely in the ovary of challenged animals (Figure 3B). Together with the lack of protection observed in animals immunized with 0.25 μg OMVs (Figure 3B), these data highlight the complexity of determining the most effective OMV dosage in immunization trials. When considering the dose/response correlation of the three OMV concentrations tested in study 2, we can see that the overall protection conferred presents a positively skewed distribution, peaking at 2.5 μg (Figure 3B). This dose/response correlation is not unlike what has been described for other protein-based vaccine formulations, where significant protection can sometimes be obtained by administering 50–1000 or 250–5000 μg of antigen [40]. Nonetheless, our data showed that the range of effective immunization concentration is certainly narrower in the case of the OMVs. This observation may be explained considering once more the adjuvant properties of the OMVs and the host immune response to the vesicles. The OMVs exhibit on their surface a variety of pathogen-associated molecular patterns (PAMPs) [13], molecules able to elicit an immune response in the host ultimately by recruiting and activating antigen-presenting cells (APCs) [20]. This process is partly mediated by the release of pro-inflammatory cytokines by host immune effectors, the concentration of which can affect the balance between an inflammatory and adaptive response [41]. Moderate levels of inflammation can potentiate the response to an administered immunogen, and for this reason pro-inflammatory compounds such as adjuvants are generally included in vaccine formulations. High levels of inflammation, such as the ones generated by the administration of high concentrations of PAMPs, are on the other hand generally disruptive for the achievement of an effective adaptive immune response [20,40]. An example of the effect of high concentrations of PAMPs on the immune system has been reported in mice, where the administration of high concentrations of LPS was shown to disrupt the B cells response to T-dependent antigens [42]. Taking this into account, we could interpret the data from study 2 as follows. 0.25 μg of OMVs likely contained an amount of PAMPs insufficient to elicit any significant immune response in the animals. Of OMVs 2.5 μg contained a sufficient amount of PAMPs able to provide the correct balance between inflammatory and adaptive response. Finally, 25 μg may have contained an excess of PAMPs, skewing the immune system more towards an inflammatory, and ultimately less effective, response. No indications of excess inflammation were however observed in relation to the sites of vaccination.

Serological data from study 2 showed that the administration of 25 μg OMVs was the most effective in eliciting IgY titers against the OMVs in the immunized animals (Figure 5B), despite the lower overall protection conferred by this dosage as compared to immunization with 2.5 μg of OMVs. The discrepancy observed in this study between IgY titers and overall protection elicited by immunization may indicate that antibody levels are often not predictive of effective protection [43,44]. In our case the development of more but less effective IgY antibodies in animals immunized with 25 μg OMVs could possibly be attributed to the hyper-inflammation caused by the administration of high levels of LPS, resulting in massive activation of T cells and disruption of the antibody maturation process [42,45].

The presence of significant IgY titers against the OMVs in the egg yolk from chickens immunized with 2.5 μg and 25 μg OMVs (Figure 5B) suggests the possibility of vertical transfer of immunity elicited in the parents by the immunization regimes adopted. Transfer of maternally derived IgYs has previously been shown to occur in chickens after vaccination against several pathogens [46,47], and is widely recognized to influence both positively [46,48] and negatively [47,49] the immunocompetence of the offspring during the early post-hatching period. In the absence of further data, we can unfortunately only speculate about the effect that passive immunity may have in protecting the offspring against *G. anatis* infections post-hatching, a hypothesis that will certainly have to be investigated in future studies.

## 5. Conclusions

Our data showed that: (i) the OMV isolation protocol based on HF represented a cost-effective and reliable method for the development of OMV-based vaccine formulations; (ii) the vaccine formulations containing OMVs alone and OMVs + FlfA + GtxA-N were the most effective in preventing morbidity associated with *G. anatis* infections; (iii) the optimal immunization concentration of OMVs for inducing protection in the immunized animals seemed to lie between 2.5 μg and 25 μg, although higher serum and yolk specific IgY titers were obtained by immunization with 25 μg and (iv) the IgY-dependent immunity induced by immunization of 2.5 μg and 25 μg OMVs could be subject to vertical transfer and possibly exert some degree of maternally derived protection in the newly hatched chicks.

## Figures and Tables

**Figure 1 vaccines-08-00040-f001:**
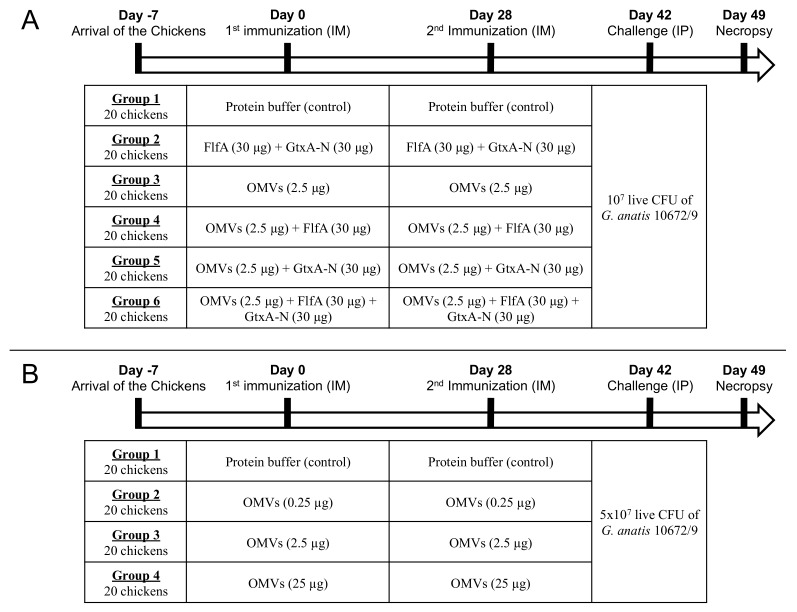
Experimental design of study 1 (**A**) and 2 (**B**). IM: intramuscular; IP: intraperitoneal; CFU: colony-forming units.

**Figure 2 vaccines-08-00040-f002:**
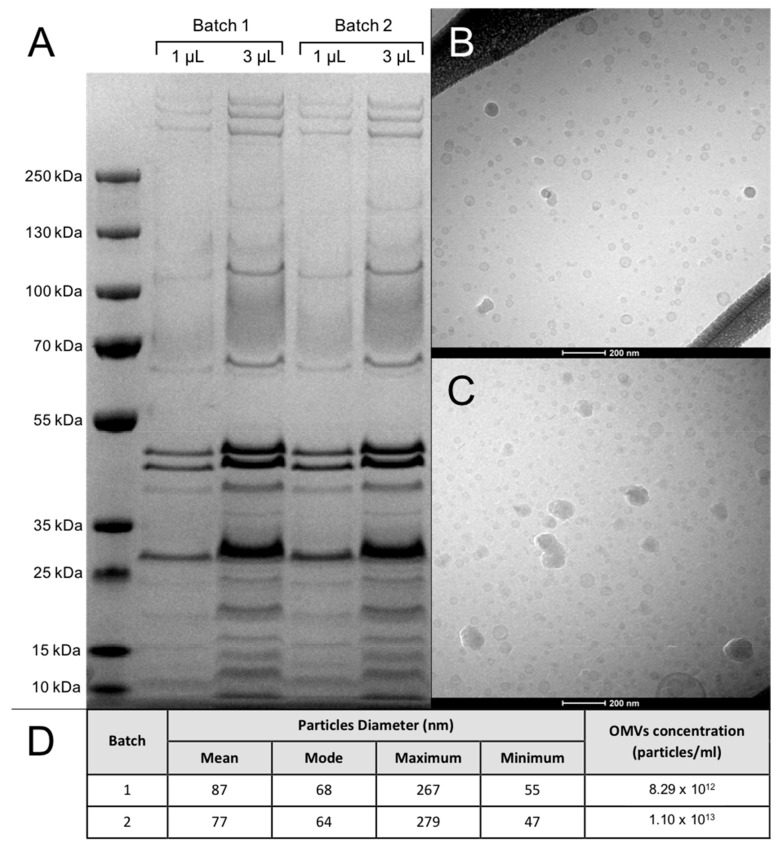
Characterization of outer membrane vesicle (OMV) batches. (**A**) SDS-PAGE analysis of OMV batches. 1: protein ladder; 2–3: OMVs batch 1; 4–5: OMVs batch 2. (**B**,**C**) Cryo-TEM analysis of OMV batches 1 (**B**) and 2 (**C**). (**D**) Tunable resistive pulse sensing (TRPS) characterization of OMV batches 1 and 2.

**Figure 3 vaccines-08-00040-f003:**
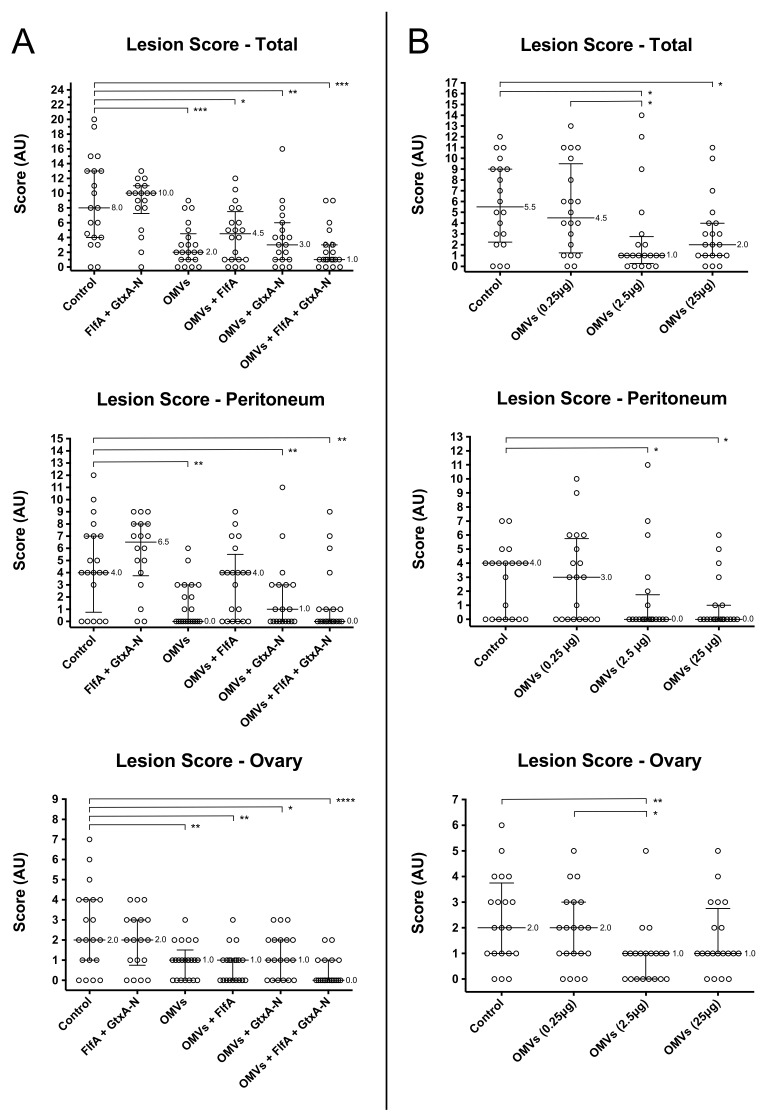
Lesion scores from study 1 (**A**) and 2 (**B**). Peritoneum (top), ovary (middle) and overall (bottom) scores are shown. Median values and interquartile ranges of the scores of each group are reported on the graphs. AU: arbitrary units. *: *p* < 0.05; **: *p* < 0.01; ***: *p* < 0.001; ****: *p* < 0.0001. *p*-values from Mann–Whitney test shown.

**Figure 4 vaccines-08-00040-f004:**
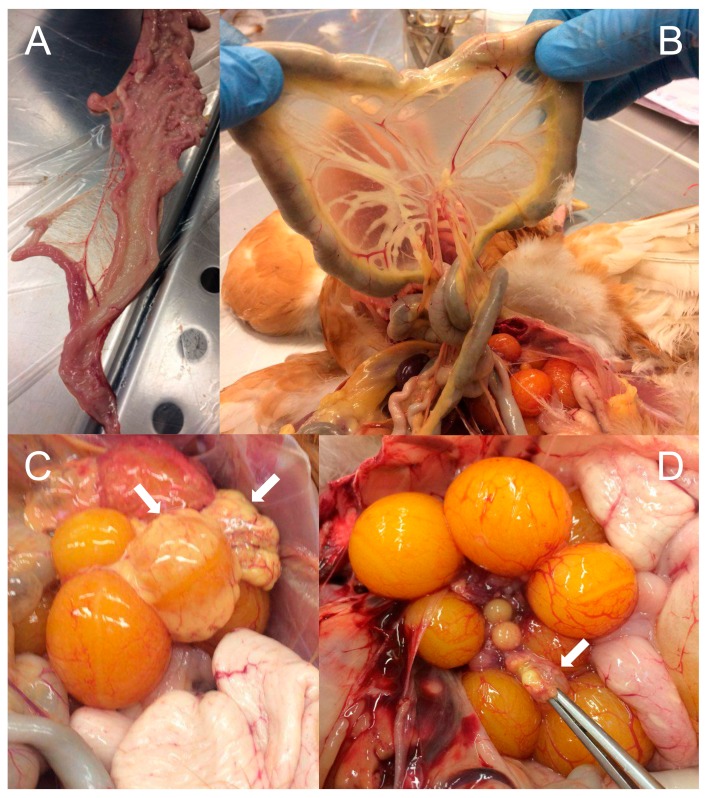
Pathological analysis of different organs during post-mortem examination (necropsy). Clinical signs typically associated with *G. anatis* infections are visible. (**A**) Purulent salpingitis. (**B**) Peritonitis. (**C**) Purulent inflammation on several follicles. (**D**) Follicle regression. White arrows indicate individual follicles.

**Figure 5 vaccines-08-00040-f005:**
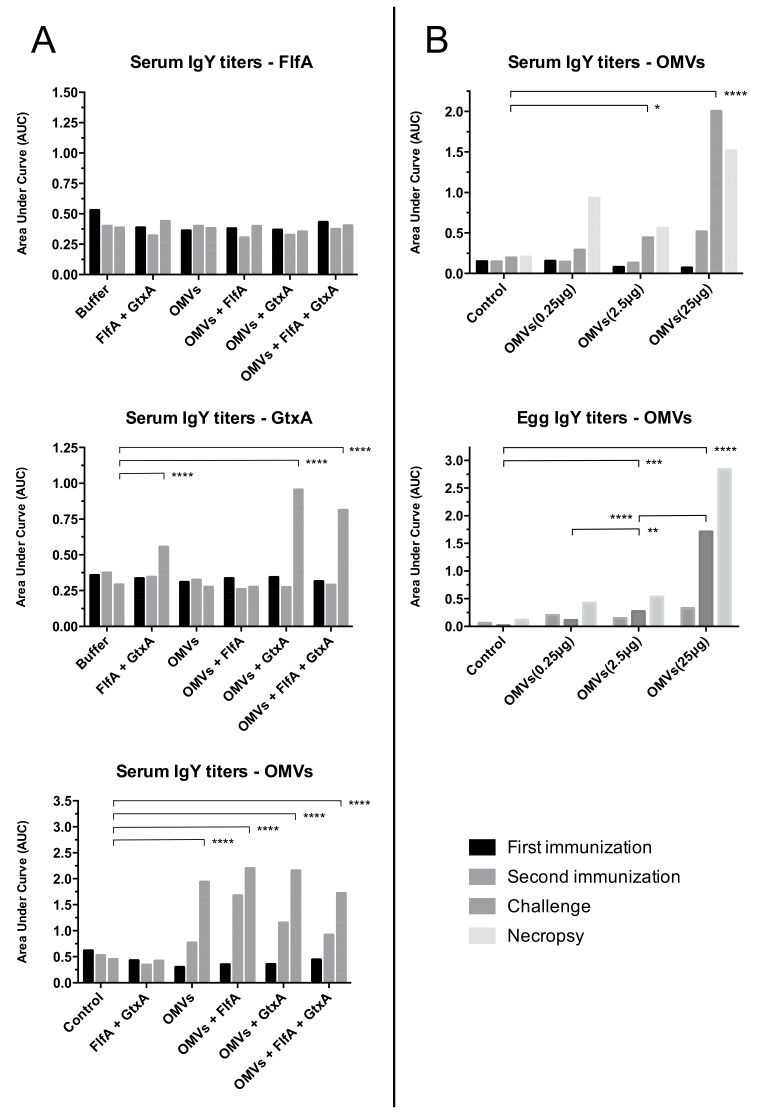
IgY response to immunization. Sera and yolk IgY from study 1 (**A**) and 2 (**B**) were pooled together within each individual group and analyzed for IgY response against FlfA, GtxA-N and *G. anatis* OMVs. Data are reported as the area under curve (AUC), calculated from ELISA titration curves. Four time points are shown: first immunization (T0); second immunization (T1); challenge (T2); necropsy (T3). *: *p* < 0.05; **: *p* < 0.01; ***: *p* < 0.001; ****: *p* < 0.0001. *p* values from Dunnett’s multiple comparison test shown (alpha = 0.05).

**Table 1 vaccines-08-00040-t001:** Colony-forming units (CFU) count from the inocula containing *Gallibacterium anatis* (10672/9).

Study	Inoculum	Administered to Group	CFU/mL (Pre-Challenge)	CFU/mL (Post-Challenge)
1	1	1, 2	0.92 × 10^7^	0.8 × 10^7^
	2	3, 4	0.8 × 10^7^	0.81 × 10^7^
	3	5, 6	1.25 × 10^7^	1.10 × 10^7^
2	1	1, 2	5.58 × 10^7^	3.83 × 10^7^
	2	3, 4	5.79 × 10^7^	4.9 × 10^7^

Serial dilutions from challenge inocula before and after challenge were plated on BHI-agar plates supplemented with 5% *v*/*v* blood. CFU counts are expressed as CFU/0.5 mL.

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
