# Peer review of "Hydrostatic Filtration Enables Large-Scale Production of Outer Membrane Vesicles That Effectively Protect Chickens against Gallibacterium anatis"

_vaccines, 2020, doi:10.3390/vaccines8010040_

Round 1
Reviewer 1 Report
The authors examined the vaccine efficacy of OMVs derived from G. anatis and their potential as an adjuvant. Then, the authors prepared OMVs by hydrostatic filtration to solve the problem concerning the cost and mass-production of vaccines. This study revealed that the immunization with OMVs indicated the protection against the diseases caused by G. anatis infection, and had a potential as an adjuvant. In addition, IgY titers against OMVs were increased in sera and egg yolk from the immunized animals.
General comments
The cost and mass-production in manufacturing vaccines are sometimes big barriers in the development of novel vaccines for the livestock and poultries. In this manuscript, the authors adopted HF for the preparation of OMVs to reduce the cost in manufacturing and for the mass production. The OMVs prepared indicated high efficacy for the protection against G. anatis infection. The utilization of OMVs prepared by HF seems to be valuable for the development of vaccines against G. anatis infection, and it seems to have a potential that OMVs could be applied to other infectious diseases in the veterinary medicine.
On the other hand, the vaccine efficacy of the combined antigens of OMVs and GtXA-N was lower than those of OMVs alone (Fig. 3A), although the antibodies specific to OMVs and GtXA-N were produced (Fig. 5A). Therefore, the combined antigens of OMVs and GtXA-N seemed to fail to induce effective antibody titer for the protection. In addition, the combination of OMVs and FlfA failed to induce the antibody specific to FlfA (Fig 5A). Thus, further experiments would be required to suggest the utility of OMVs as an adjuvant.
Minor comments
Line 90: heart?
Line 144: ARRIVE?
Lines 158-159: How did the authors prepare the antigens combined with OMVs? Was it simply mixed?
Line 229: “particle size and number” is seemed to be indicated in Fig. 2D. Please confirm it.
Figure 2D: Improve character corruptions.
Table 1: Lines should be added in upper and lower ends of this table.
Line 308: Are the components in the OMVs from delta-tolR mutant similar to those in wild type bacteria?
Lines 395-397: Do the authors have any data indicating the expression levels of inflammatory cytokines? Or, were any clinical signs indicating hyper-inflammation observed in the immunized chickens?
Author Response
Reviewer 1:
The authors examined the vaccine efficacy of OMVs derived from G. anatis and their potential as an adjuvant. Then, the authors prepared OMVs by hydrostatic filtration to solve the problem concerning the cost and mass-production of vaccines. This study revealed that the immunization with OMVs indicated the protection against the diseases caused by G. anatis infection, and had a potential as an adjuvant. In addition, IgY titers against OMVs were increased in sera and egg yolk from the immunized animals.
General comments
The cost and mass-production in manufacturing vaccines are sometimes big barriers in the development of novel vaccines for the livestock and poultries. In this manuscript, the authors adopted HF for the preparation of OMVs to reduce the cost in manufacturing and for the mass production. The OMVs prepared indicated high efficacy for the protection against G. anatis infection. The utilization of OMVs prepared by HF seems to be valuable for the development of vaccines against G. anatis infection, and it seems to have a potential that OMVs could be applied to other infectious diseases in the veterinary medicine.
Correct.
On the other hand, the vaccine efficacy of the combined antigens of OMVs and GtXA-N was lower than those of OMVs alone (Fig. 3A), although the antibodies specific to OMVs and GtXA-N were produced (Fig. 5A). Therefore, the combined antigens of OMVs and GtXA-N seemed to fail to induce effective antibody titer for the protection. In addition, the combination of OMVs and FlfA failed to induce the antibody specific to FlfA (Fig 5A). Thus, further experiments would be required to suggest the utility of OMVs as an adjuvant.
We partly disagree to this point. It is correct that the median lesion score was lower for the OMV vaccinated only group (2) compared to the OMV plus GtxA-N vaccinates (3). Yet, there was no statistical difference between the total lesion scores of any of the vaccine groups where OMVs were included. In other words no additive protective effect was demonstrated from combining OMVs with either FlfA or GtxA-N on the total lesion score. However, when comparing to the non-vaccinated controls we believe there are indications of an additive effect particularly when assessing lesions in the individual organ systems e.g. the ovary. Here the combination OMV + FlfA + GtxA-N result in highly significant p-values (p<0.0001) as opposed to OMVs alone (p<0.01). We do not make claims regarding the adjuvant effect of OMVs in the paper although we believe an adjuvant effect can be expected at least under some circumstances. Based on these reflections we have not changed the text in the manuscript.
Minor comments
Line 90: heart?
True. The spelling error has been corrected.
Line 144: ARRIVE?
True. The spelling error has been corrected.
Lines 158-159: How did the authors prepare the antigens combined with OMVs? Was it simply mixed?
Yes, the protein antigens were prepared by simply mixing the OMVs and the respective proteins in the protein buffer, as stated in the materials and methods section.
Line 229: “particle size and number” is seemed to be indicated in Fig. 2D. Please confirm it.
Correct. The mean particle size in the two batches was 82 nm and the number of OMVs was 9.65 x 1012 OMV particles per ml.
Figure 2D: Improve character corruptions.
Not understood. Please indicate which characters are considered corrupted. The number of particles is reported as 8.29e+12, which is another way of writing 8.29 x 1012. We suggest to leave the writing as is.
Table 1: Lines should be added in upper and lower ends of this table.
The suggested lines have been added.
Line 308: Are the components in the OMVs from delta-tolR mutant similar to those in wild type bacteria?
Yes. Although we have not made an exhaustive proteomic characterization of the OMVs from the wild type and the delta-tolR, we have previously reported that the protein profiles are very similar and only seem to differ in quantity for some proteins. Further details can be obtained in Bager, R.J.; Nesta, B.; Pors, S.E.; Soriani, M.; Serino, L.; Boyce, J.D.; Adler, B.; Bojesen, A.M. The fimbrial protein FlfA from Gallibacterium anatis is a virulence factor and vaccine candidate. Infect. Immun. 2013, 81, 1964–1973, which is cited in the text. The point as been clarified further in the discussion (Lines 309-311)
Lines 395-397: Do the authors have any data indicating the expression levels of inflammatory cytokines? Or, were any clinical signs indicating hyper-inflammation observed in the immunized chickens?
No, we did not characterize expression levels of inflammatory cytokines nor did we observe clinical signs suggesting hyper-inflammation in the immunized birds in relation to the immunizations. This has been mentioned in the discussion (Lines 390-391).
Reviewer 2 Report
In this manuscript, Antenucci et al., describe a new way of production of outer membrane vesicles of Gallibacterium anatis. This manuscript shows how this kind of production can enhance the vaccine area by increasing the cost-effective large scale production.
They also compare the different combinations tested in earlier publications, with FlFa and GtxA-N proteins, and the adequate dose for vaccination purposes.
The manuscript is well written, and the experiments follow a logic rationale.
I have some comments/suggestions for the authors:
What is the rationale of using 30 micrograms of each? figure 1. Authors should explain why the use the G anatis strain 10672/9. It would be interesting to compare the same results with OMVs isolated by other ways such as ultracentrifugation, etc. What is the specificity of theses OMVs in the protection against G anatis?could the authors use other OMVs from other bacteria as a negative control?
Author Response
Reviewer 2:
In this manuscript, Antenucci et al., describe a new way of production of outer membrane vesicles of Gallibacterium anatis. This manuscript shows how this kind of production can enhance the vaccine area by increasing the cost-effective large scale production.
They also compare the different combinations tested in earlier publications, with FlFa and GtxA-N proteins, and the adequate dose for vaccination purposes.
The manuscript is well written, and the experiments follow a logic rationale.
Thank you for the supportive comments.
I have some comments/suggestions for the authors:
What is the rationale of using 30 micrograms of each? figure 1.
Determining the most effective protein antigen dose is not trivial and potentially requires a large number of animals. We have previously used 50 ug to 100 ug for immunizations with FlfA and GtxA-N in chickens and generally observed good protective effects and antibody productions. In the current investigation we aimed at using a slightly lower antigen amount to save material and lower cost. This did however not appear to affect the protective capacity in term s of lesion scores. On the other hand the lower antigen concentration of FlFA may have affected the ability to raise FlfA specific antibodies. This will however have to be investigated further in a future study. As the amount of antigen is within normal limits used in for this purpose we have not included further comments on this in the paper.
Authors should explain why the use the G anatis strain 10672/9.
The OMVs used to immunize chickens in the current investigation originate from the G. anatis strain 12656-12. To document the breath of protection we challenged the immunized birds with the G. anatis strain 10672/9, which belongs to a different serotype than strains 12656-12. In other words, we aimed at a heterologous challenge to test the breath of protection. This aspect has been clarified in the Materials and Methods section (Lines 168-170).
It would be interesting to compare the same results with OMVs isolated by other ways such as ultracentrifugation, etc.
Agree. This has been done, at least partly, and reported in the manuscripts below:
Bager, R.J.; Nesta, B.; Pors, S.E.; Soriani, M.; Serino, L.; Boyce, J.D.; Adler, B.; Bojesen, A.M. The fimbrial protein FlfA from Gallibacterium anatis is a virulence factor and vaccine candidate. Infect. Immun. 2013, 81, 1964–1973.
Pedersen, I.J.; Persson, G.; Bojesen, A.M.; Pors, S.E.; Skjerning, R.B.; Thøfner, I.C.N. Outer membrane vesicles of Gallibacterium anatis induce protective immunity in egg-laying hens. Vet. Microbiol. 2016, 195, 123–127.
Both papers are already cited in the current manuscript.
What is the specificity of theses OMVs in the protection against G anatis?
We have not made an exhaustive investigation of the protective breath of OMV from one strain of G. anatis against a broad range of other G. anatis strains. Yet, by challenging immunized bird with a so called heterologous strain (different serotype), we have observed some indications of the protective breath. Further data to account for this question will have to be obtained in a future investigation. As determination of the protective breath was not a the focus of the current investigation we have not included specific points on this in the paper.
Could the authors use other OMVs from other bacteria as a negative control?
OMVs are complex entities containing several immunogenic components some of which may share conformational and linear epitopes with antigens present in distantly related bacteria. Due to that it is not trivial to select another bacterial species or strain as a negative control as they may at least partly induce cross protective immunity. One of the advantages of using OMVs as a vaccine, when comparing to recombinant proteins, is in fact that they contain a complex mix of immunogens, typically in their natural conformation, that resembles the parent strain. This perspective has not been added to the discussion as we think this aspect would need specific data for not being too speculative. The points has thus not ben addressed in the manuscript.
Round 2
Reviewer 1 Report
The manuscript has been improved, and the authors have addressed my concerns.
Minor comments
Line 229: “particle size and number (Fig. 2C).” should be modified to “particle size and number (Fig. 2D).”, as commented in the authors’ reply.
Figure 2D: Improve character corruptions. I attached the Fig2D. It may be a problem of the conversion system to build a PDF file. Many “?” marks are found. Confirm it.

Author Response
Line 229: “particle size and number (Fig. 2C).” should be modified to “particle size and number (Fig. 2D).”, as commented in the authors’ reply.
The requested change has been made in the manuscript.
Figure 2D: Improve character corruptions. I attached the Fig2D. It may be a problem of the conversion system to build a PDF file. Many “?” marks are found. Confirm it.
We see the problem now but couldn't see it in the previous version. A revised version of Fig 2 has been inserted in the manuscript file uploaded. We hope the problem with the conversion has been solved.